# Intermediate Representations are Strong AI-Generated Image Detectors

## Abstract

The rapid advancement in generative AI models has enabled the creation of photo-realistic images. At the same time, there are growing concerns about the potential misuse and dangers of generated content, as well as a pressing need for effective AI-generated image detectors. However, current training-based detection techniques are typically computationally costly and can hardly be generalized to unseen data domains, while training-free methods fall short in detection performance. To bridge this gap, we propose a search-based method employing data embedding sensitivity in intermediate layers to detect AI-generated images. Given a set of real and AI-generated images, our method scans through the detection performance in the composite configuration space of intermediate layer, perturbation type, and severity level to identify the best configuration for detection. We examine the proposed method on two comprehensive benchmarks: GenImage and DF40. Our method exhibits improved performance across different datasets compared to both training-free and training-based state-of-the-art methods. On average, our method outperforms the best training-free/training-based methods on the GenImage benchmark by 16.1%/4.9% and on the DF40 benchmark by 14.5%/8.7% in AUROC score. We release the code at `https://anonymous.4open.science/r/Intermediate-Public-D256`.

## 1 Introduction

The advent of image generative models enables the creation of realistic synthetic images. Fueled by advances in deep learning techniques, generative models such as generative adversarial network (GAN) (Goodfellow et al., 2020; Metz et al., 2016; Liu & Tuzel, 2016; Mao et al., 2017; Yoon et al., 2019; Karras et al., 2019), Variational Autoencoder (VAE) (Mescheder et al., 2017; Mishra et al., 2018; Pinheiro Cinelli et al., 2021; He et al., 2022), diffusion model (Ho et al., 2020; Song et al., 2020; Saharia et al., 2022; Podell et al., 2023; Blattmann et al., 2023; Peebles & Xie, 2023), *etc.* have demonstrated significant progress in image generation. While some image-generation applications have attracted users to go bananas, generative models pose serious ethical, societal, and security challenges. The misuse and the associated cost of generated images can cause negative impacts such as copyright violation, deepfake, and fake content in publications. Furthermore, training datasets for deep learning models might be corrupted by generated images at scale, leading to unintentional bias or malicious exploits for future models. These critical challenges underscore the need for reliable AI-generated image detection.

There are two mainstream approaches to detecting AI-generated images: *training-based* and *training-free* approaches. Current training-based approaches have limited generalization to unseen data domains, while training-free approaches have inferior detection performance. To bridge the gap, we propose a simple yet effective training-free detector that exploits a pre-trained image foundation model to detect AI-generated images. Following prior arts in training-detection (He et al., 2024; Tsai et al., 2024) that use a similarity score computed by a pair of test image and its perturbed version for detection, our method firstly considers the exploration of the best *configuration* to derive the most discriminative feature between real and AI-generated images, where the space of configurations is a tuple consisting of (i) the layer index of the model, (ii) the perturbation type, and (iii) the severity level of the selected perturbation type. Given a set of real and AI-generated images, our method calculates the similarity scores across all configurations and selects the optimal one for detection. For example, our implementation uses CLIP (Radford et al., 2021) (ViT-L/14 image encoder) as

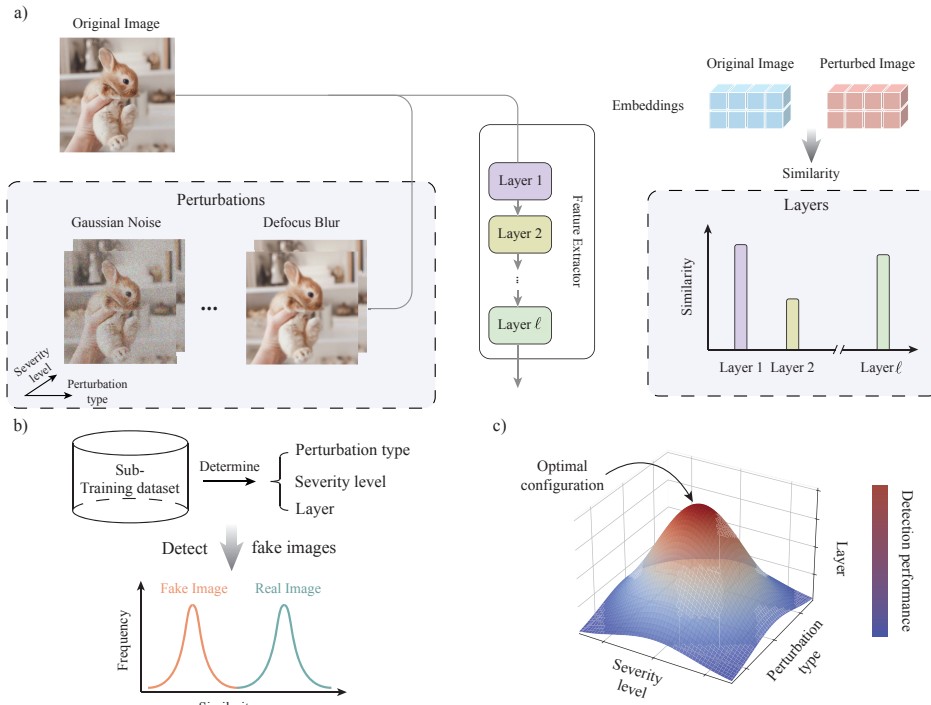

Figure 1: Illustration of the proposed method. (a) Both the original image and the perturbed image are fed to the feature extractor (a pretrained image foundation model). Embeddings across all layers are extracted to obtain intermediate representations. The cosine similarity between the embeddings of the original image and the perturbed image is computed as the metric to make a binary classification on whether an image is AI-generated. (b) We use a small portion of the training dataset to determine which perturbation (including perturbation type and severity level) and embedding from which layer are best to be used to compute the similarity for detection. (c) The configuration search space includes a combination of the optimal intermediate layer, perturbation type, and severity level.

the backbone model (with 25 layers) together with 8 unique image perturbation functions and 8 different severity levels. This yields a total of $25 \times 8 \times 8 = 1600$ configurations. Consequently, some training-free methods such as (He et al., 2024; Tsai et al., 2024) can be viewed as a special case of our method with a fixed configuration that only leverages the embedding from a designated layer and considers a limited set of perturbation types (usually less than two). Figure 1 illustrates the overview of our proposed search-based detector. By scaling up the configuration space, our method exhibits better performance compared to both training-free approaches and training-based approaches on the GenImage benchmark (Zhu et al., 2023) and the DF40 benchmark (Yan et al., 2024b).

## 2 RELATED WORK

**AI-Generated Image Detection**   Frequency domain analysis is found to be effective to detect AI-generated images (Frank et al., 2020; Chandrasegaran et al., 2021; Corvi et al., 2023a). In addition to handcrafted features, learning-based methods are proposed to exploit the strength of neural networks (Corvi et al., 2023b; Cozzolino et al., 2021; Gragnaniello et al., 2021; Ojha et al., 2023). UniDetector (Ojha et al., 2023) uses both nearest neighbor (training-free) and linear probing (training-based) on the image embedding space to detect AI-generated images. NPR (Tan et al., 2024) trains a detector that is generalizable to detect images generated by both GANs and diffusion models. The detector relies on neighboring pixel relationships based on the observation that local independence among image pixels exhibits generalized forgery artifacts in generated images. AIDE (Yan et al., 2024a) captures both low-level pixel statistics and high-level global semantics to detect anomalies in AI-generated images such as white noise in the image (low level) and unreasonable image components in the context (high level). SPAI (Karageorgiou et al., 2025) uses spectral learning to distinguish AI-generated images based on the spectral reconstruction similarity.

In addition to learning-based methods, training-free methods, not limited to the training dataset, are proposed. AeroBlade (Ricker et al., 2024) assumes that the reconstruction of AI-generated images is easier than that of real images. Hence, the reconstruction error can be used as the metric to detect AI-generated images. RIGID (He et al., 2024) assumes that AI-generated images are less robust to perturbations in the embedding space of neural architectures. MINDER (Tsai et al., 2024) improves the prediction of the RIGID method by introducing contrastive perturbation.

**Exploiting Intermediate Layers** Intermediate layers are found to be able to enhance the prediction and assist in the analysis of neural architectures. They are used to predict generalization gaps (Jiang et al., 2018), elucidate training dynamics through linear classifier probes (Alain & Bengio, 2016), improve transfer learning (Evci et al., 2022), enhance the adversarial example transferability (Huang et al., 2019), and ameliorate the performance of fine-tuned models (Lee et al., 2022). A fundamental geometric property of the data representation in over-parameterized neural networks is the *intrinsic dimension*, *i.e.* the minimal number of coordinates necessary to describe data points without significant information loss. It is found that the intrinsic dimension increases in earlier layers (expansion) and decreases in later layers (compression) (Ansuini et al., 2019; Recanatesi et al., 2019).

## 3 INTERMEDIATE REPRESENTATIONS AS AI-GENERATED IMAGE DETECTORS

The overall flow of this section is as follows: First, we formally define the task formulation of our search-based detection framework. Then, we present our proposed method and the algorithm. Next, we provide motivating examples to articulate the importance of selecting the right layer to obtain discriminative features for detection. Finally, we explain why intermediate representations are powerful features for AI-generated image detection through the lens of intrinsic dimension analysis.

**Task Formulation** Given a set of labeled images $\mathcal{D} = \{(\mathbf{x}_i, y_i)\}_{i=1}^n$ with $\mathbf{x}_i \in \mathcal{X}$ denoting an image and $y_i \in \{0, 1\}$ denoting its label. $y_i = 1$ indicates AI-generated image while $y_i = 0$ indicates real image. Using a pretrained image feature extractor $\mathcal{F}(\cdot)$, the goal is to assign a predicted label $\hat{y}$ for a test image $\mathbf{x}$. The aim of this paper is to explore the potential of intermediate representations for search-based AI-generated image detection. This will be accomplished by studying the effect of expanding the configuration search space (see Figure 1 (c)), which consists of the intermediate layers of $\mathcal{F}(\cdot)$, perturbation types, and severity levels. The search-based detection process does not modify the weights or structure of the pretrained image feature extractor. Labeled images are used only to determine the optimal configuration in the search space.

### 3.1 PROPOSED METHOD

Figure 1 shows the illustration of the proposed method. We feed both the original image $\mathbf{x}$ and the perturbed image $\epsilon(\mathbf{x})$ to the model $\mathcal{F} = f_L \circ \ldots f_\ell \ldots \circ f_1$, where $f_\ell$ denotes the $\ell$-th layer of $\mathcal{F}$. Both $\mathbf{x}$ and $\epsilon(\mathbf{x})$ constitute a pair to compute the cosine similarity that characterizes the drift in the embedding space caused by a perturbation. Eight perturbation types and eight severity levels are applied. Perturbation types include Gaussian noise, shot noise, impulse noise, defocus blur, zoom blur, contrast, elastic transform and JPEG compression. Those perturbations are algorithmically generated corruptions following (Hendrycks & Dietterich, 2019). Details on perturbations are reported in Appendix B. For each perturbation type, a severity level is used to control the level of corruption on $\mathbf{x}$. We use $\epsilon(\mathbf{x}|s)$ to denote the perturbed version of $\mathbf{x}$ under the perturbation $\epsilon(\cdot|s)$ with a severity level of $s$. We extract embeddings in the $l$-th intermediate layer $\mathcal{F}_{\text{sub}} = f_l \circ \ldots f_1$, $1 \leq l \leq L$, and compute the cosine similarity between the embeddings of the original image and the perturbed image. Let $\text{emb}(\cdot)$ denote the function to extract the class embedding $\mathbf{E}_l \in \mathbb{R}^d$ as the intermediate representation for each layer. For example, in DINOv2 and CLIP, $\text{emb}(\cdot)$ extracts [CLASS] token embedding. The cosine similarity of given a configuration tuple $(\epsilon, s, l)$ is defined as

$$S(\mathbf{x}, \epsilon(\mathbf{x}|s), l) = \text{sim}\big(\text{emb}(f_l \circ \ldots \circ f_1(\mathbf{x})), \text{emb}(f_l \circ \ldots \circ f_1(\epsilon(\mathbf{x}|s)))\big),$$

$$\text{sim}(\mathbf{v}_1, \mathbf{v}_2) = \frac{\langle \mathbf{v}_1, \mathbf{v}_2 \rangle}{\|\mathbf{v}_1\| \|\mathbf{v}_2\|}, \tag{1}$$

where $\langle \cdot, \cdot \rangle$ denotes the inner product of two vectors, and $\| \cdot \|$ is the Euclidean norm. $d$ is the hidden dimension defined in the feature extractor.

The label prediction for an input image is a threshold-based approach defined as

$$\hat{y} = \psi(\mathbb{I}\{S(\mathbf{x}, \epsilon(\mathbf{x}|s), l) \leq \tau\}), \tag{2}$$

where $\tau$ is a threshold to distinguish AI-generated and real images. $\mathbb{I}\{\cdot\}$ is the indicator function and $\mathbb{I}\{\mathcal{A}\} = 1$ if and only if an event $\mathcal{A}$ happens. $\psi(\cdot)$ indicates the relative robustness to perturbations, and is determined by the training dataset. Given a configuration, if real images exhibit higher similarity than AI-generated ones in the embedding space, then $\psi(x) = x$. Otherwise, $\psi(x) = 1 - x$.

Algorithm 1 depicts the pipeline for detecting AI-generated images. There are two stages: in stage I, we determine the optimal configuration using a subset of the training dataset. The best configuration is selected based on the Area Under the Receiver Operating Characteristic Curve (AUROC) score, and it comprises the optimal intermediate layer, perturbation type, and severity level. We empirically find that only a small portion of the training dataset (by default, we use 30% of the test dataset size) is sufficient to deliver stable detection performance. In stage II, a test image undergoes detection using the best configuration selected by stage I.

---

**Algorithm 1** Using intermediate representations to detect AI-generated images

---

**Require:** Randomly sampled training dataset $\mathcal{D}_{\text{tr}} = \{(\tilde{\mathbf{x}}_i, \tilde{y}_i)\}_{i=1}^{N_{\text{tr}}}$, a test image $\mathbf{x}$, a pretrained foundation model $\mathcal{F} = f_L \circ \ldots \circ f_1$, $M$ perturbation types, and $S$ severity levels
1: # Stage I: determine the best configuration
2: Initialize an empty list $\mathcal{P} \leftarrow \{\}$.
3: **for** $i = 1$ to $N_{\text{tr}}$ **do**
4:     **for** $\epsilon \in \{\epsilon_1, \ldots, \epsilon_M\}$ **do**                 ▷ Iterate over different perturbation types
5:         **for** $s \in \{1, \ldots, S\}$ **do**               ▷ Iterate over different perturbation levels
6:             $\hat{p} \leftarrow S(\tilde{\mathbf{x}}_i, \epsilon(\tilde{\mathbf{x}}_i|s), l)$ as shown in Equation 1      ▷ Compute cosine similarity
7:             $\mathcal{P} \leftarrow \mathcal{P} \cup \{\hat{p}\}$
8:         **end for**
9:     **end for**
10: **end for**
11: $(\epsilon_*(\cdot|s_*), l_*) \leftarrow \underset{\epsilon, s, l}{\arg\max} \, \text{AUROC}(\mathcal{P}, \{\tilde{y}_i\})$
12: # Stage II: inference with the best configuration
13: Make a prediction using $\mathbf{x}$, $\epsilon_*(\mathbf{x}|s_*)$ and $l_*$ as shown in Equation 2

---

## 3.2 REVISITING IMAGE EMBEDDINGS FOR AI-GENERATED IMAGE DETECTION

Prior training-free methods, such as RIGID (He et al., 2024) and MINDER (Tsai et al., 2024), postulate that AI-generated images are less robust than real images in the embedding space. We empirically find that this postulation holds true in most cases. However, there are exceptions. For example, in Figure 2, we calculate the average of cosine similarity between original and perturbed embeddings in different layers for AI-generated and real images, respectively. The DDIM dataset in the DF40 benchmark reveals that real images are less robust compared to AI-generated images. Exceptions are not limited to the feature extractor we use, *i.e.* CLIP image encoder. Other models such as DINOv2 also exhibit exceptions of robustness in the embedding space (details are reported in Appendix B.2). The result indicates that the postulation might require scrutiny. Hence, in our proposed method, we eliminate the assumption that the embeddings of real images are more robust than those of AI-generated images. In other words, the former might not necessarily have higher cosine similarity between original and perturbed embeddings than the latter. We design the $\psi(\cdot)$ function in Equation 2 to capture the relative robustness for real and AI-generated images to a perturbation. In addition, different layers exhibit different sensitivity to a perturbation, which motivates us to pursue an optimal intermediate layer to detect AI-generated images.

It is worth noting that both RIGID and MINDER focus on limited perturbation types: only Gaussian noise and Gaussian blur are considered. To give a comprehensive examination of intermediate representations as features, we use eight different perturbation types and eight severity levels, including Gaussian noise, shot noise, impulse noise, defocus blur, zoom blur, contrast, elastic transform and JPEG compression. Details on various perturbation types are reported in Appendix B.

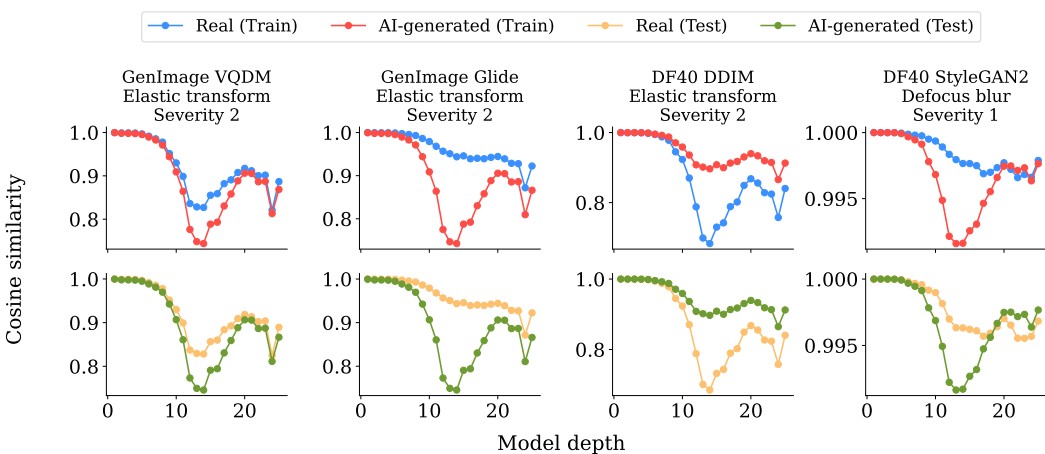

Figure 2: Average cosine similarity profile over model depth. We randomly sampled images in the train dataset with a size of 30% test dataset size to represent the training dataset in the plot. We use the CLIP model (ViT-L/14) as the feature extractor.

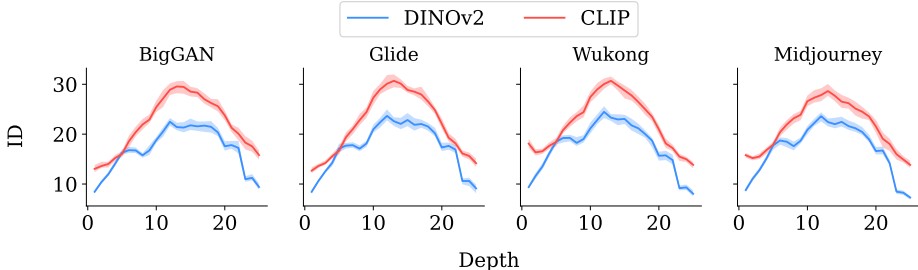

Figure 3: Intrinsic Dimension (ID) analysis of data representation manifolds in the image foundation models: DINOv2 (Oquab et al., 2023) and CLIP (ViT-L/14) (Radford et al., 2021). A typical hunchback shape of the profile of the intrinsic dimension is observed, which indicates more diverse features in intermediate layers.

## 3.3 UNDERSTANDING THE VERSATILITY OF INTERMEDIATE REPRESENTATIONS VIA INTRINSIC DIMENSION

Intrinsic dimension (ID) is a fundamental geometric property of the data representation manifold in an over-parameterized neural network. It represents the minimal number of coordinates to describe data points without significant information loss. In the learning theory, ID plays a vital role in learning function approximations and non-linear decision boundary determination. The number of required data points grows exponentially with the manifold's ID for learning a manifold (Narayanan & Mitter, 2010). ID is found to be correlated with adversarial training of neural networks (Ma et al., 2018; Amsaleg et al., 2017). A theoretical analysis indicates that an increase in ID effectively reduces the severity level of the perturbation to move a normal example into the adversarial region (Amsaleg et al., 2017). By employing ID estimator (Facco et al., 2017; Ansuini et al., 2019), we examine ID across layers of the feature extractor. ID is calculated based on the ratio between the distances to the second and first nearest neighbor of each data point (Facco et al., 2017). Figure 3 shows the variation of ID for feature extractors used in this study. There is ID expansion in earlier layers and compression in later layers. The hunchback shape of ID as a function of model depth is interpreted as the feature generation in earlier layers (Olshausen & Field, 1997; Babadi & Sompolinsky, 2014) and feature selection in later layers (Hinton & Salakhutdinov, 2006; Tishby, 2018).

The dimensionality analysis indicates that there are more diverse features in intermediate layers than in output layers. Different layers can have different levels of sensitivity to a perturbation. The output layer might not be the most sensitive layer, rendering it sub-optimal in detecting AI-generated images. As shown in Figure 2, there is a pronounced variation in cosine similarity across model layers. It indicates that different model layers might have different sensitivity to a perturbation. Besides, the

Table 1: Comparison of AUROC and AP scores on the GenImage benchmark (Zhu et al., 2023). Ours (CC) uses a consistent configuration across the benchmark.

| Method | Metric | BigGAN | SD v4 | VQDM | ADM | Glide | Midjourney | SD v5 | Wukong | Avg |
|---|---|---|---|---|---|---|---|---|---|---|
| | | | | | *Training-free method* | | | | | |
| AeroBlade | AUROC | 0.9352 | 0.6287 | 0.8965 | 0.8371 | 0.8207 | 0.7128 | 0.5342 | 0.6134 | 0.7473 |
| | AP | 0.9013 | 0.6034 | 0.9060 | 0.8227 | 0.8211 | 0.6974 | 0.5137 | 0.6078 | 0.7342 |
| RIGID | AUROC | 0.9882 | 0.6508 | 0.9390 | 0.9146 | 0.9779 | 0.7422 | 0.6502 | 0.6391 | 0.8128 |
| | AP | 0.9860 | 0.6230 | 0.9424 | 0.9162 | 0.9774 | 0.7244 | 0.6295 | 0.6154 | 0.8018 |
| MINDER | AUROC | 0.9270 | 0.6579 | 0.9377 | 0.8919 | 0.8372 | 0.7386 | 0.6568 | 0.6482 | 0.7869 |
| | AP | 0.9156 | 0.6360 | 0.9412 | 0.8885 | 0.8372 | 0.7149 | 0.6423 | 0.6347 | 0.7763 |
| | | | | | *Training-based method* | | | | | |
| UniDetector | AUROC | 0.9700 | 0.7346 | 0.9412 | 0.8707 | 0.7870 | 0.5147 | 0.7285 | 0.8103 | 0.7946 |
| | AP | 0.9613 | 0.7007 | 0.9423 | 0.8631 | 0.7756 | 0.5164 | 0.6905 | 0.7942 | 0.7805 |
| NPR | AUROC | 0.9642 | 0.8944 | 0.8691 | 0.8430 | 0.9388 | 0.8069 | 0.8996 | 0.7901 | 0.8758 |
| | AP | 0.9585 | 0.8947 | 0.8508 | 0.8499 | 0.9488 | 0.8146 | 0.9005 | 0.7965 | 0.8768 |
| AIDE | AUROC | 0.9811 | 0.8292 | 0.9721 | 0.9639 | 0.9826 | 0.8373 | 0.8329 | 0.7949 | 0.8992 |
| | AP | 0.9836 | 0.8308 | 0.9797 | 0.9697 | 0.9887 | 0.8684 | 0.8382 | 0.7966 | 0.9070 |
| SPAI | AUROC | 0.8710 | 0.6467 | 0.6823 | 0.7005 | 0.8858 | 0.5424 | 0.6379 | 0.7074 | 0.7093 |
| | AP | 0.8735 | 0.6005 | 0.6858 | 0.6891 | 0.8873 | 0.5269 | 0.5959 | 0.6533 | 0.6890 |
| | | | | | *Search-based method* | | | | | |
| Ours | AUROC | 0.9982 | 0.9240 | 0.9475 | 0.9825 | 0.9996 | 0.9031 | 0.9209 | 0.8739 | 0.9437 |
| | AP | 0.9980 | 0.9118 | 0.9289 | 0.9813 | 0.9996 | 0.9136 | 0.9081 | 0.8569 | 0.9373 |
| Ours (CC) | AUROC | 0.9985 | 0.9257 | 0.9609 | 0.9933 | 0.9990 | 0.8303 | 0.9294 | 0.7997 | 0.9296 |
| | AP | 0.9973 | 0.8883 | 0.9278 | 0.9849 | 0.9975 | 0.8083 | 0.8920 | 0.7659 | 0.9078 |

Table 2: Comparison of AUROC and AP scores on the Forensic Small benchmark (Park & Owens, 2025). Ours (CC) uses a consistent configuration across the benchmark.

| Method | GAN | | LatDiff | | PixDiff | | Other | | Comb | |
|---|---|---|---|---|---|---|---|---|---|---|
| | AUROC | AP | AUROC | AP | AUROC | AP | AUROC | AP | AUROC | AP |
| | | | | | *Training-free method* | | | | | |
| AeroBlade | 0.4582 | 0.4623 | 0.4171 | 0.4072 | 0.6738 | 0.6037 | 0.5163 | 0.5221 | 0.4213 | 0.4176 |
| RIGID | 0.7102 | 0.7274 | 0.5362 | 0.5493 | 0.8241 | 0.8602 | 0.8687 | 0.8719 | 0.6106 | 0.6240 |
| MINDER | 0.7000 | 0.7131 | 0.5275 | 0.5413 | 0.6532 | 0.6559 | 0.8355 | 0.8367 | 0.5944 | 0.5995 |
| | | | | | *Training-based method* | | | | | |
| UniDetector | 0.8824 | 0.9024 | 0.6417 | 0.6868 | 0.7489 | 0.7375 | 0.9210 | 0.9191 | 0.7879 | 0.7023 |
| NPR | 0.8581 | 0.8494 | 0.7825 | 0.7851 | 0.8834 | 0.9017 | 0.9025 | 0.9003 | 0.8192 | 0.7669 |
| AIDE | 0.8040 | 0.8052 | 0.7500 | 0.7305 | 0.9113 | 0.8810 | 0.9455 | 0.9270 | 0.7747 | 0.8034 |
| SPAI | 0.6265 | 0.6170 | 0.6910 | 0.6799 | 0.8126 | 0.8200 | 0.6686 | 0.6761 | 0.6643 | 0.4761 |
| | | | | | *Search-based method* | | | | | |
| Ours (CC) | 0.8801 | 0.8838 | 0.8655 | 0.8663 | 0.9149 | 0.9415 | 0.9539 | 0.9403 | 0.8726 | 0.9123 |

largest difference in cosine similarity between real and AI-generated image embeddings occurs in intermediate layers.

The variation of cosine similarity in the randomly sampled training dataset follows a highly similar trend to that in the test dataset. Hence, we can use the training dataset as the prior knowledge to determine the optimal setting, including the intermediate layer, for detecting AI-generated images.

## 4 EXPERIMENTS

### 4.1 EXPERIMENTAL DETAILS

**Datasets** We evaluate the proposed method on three deepfake benchmarks: GenImage (Zhu et al., 2023), DF40 (Yan et al., 2024b) and Forensic Small (Park & Owens, 2025). GenImage consists of a broad range of image classes generated by advanced image generators, including BigGAN (Brock et al., 2018), Stable Diffusion v1.4 and v1.5 (Rombach et al., 2022), VQDM (Gu et al., 2022), GLIDE (Nichol et al., 2021), ADM (Dhariwal & Nichol, 2021), Midjourney (Midjourney, 2022) and Wukong (Wukong, 2022). The DF40 benchmark contains real images from Celeb-DF (CDF) (Li et al., 2020), FFHQ (Karras et al., 2019) and CelebA (Liu et al., 2018), as well as AI-generated images by deepfake generation techniques. Models used to yield AI-generated images include DDIM (Song et al., 2020), SiT (Ma et al., 2024), StyleGAN2 (Karras et al., 2020), StyleGAN3 (Karras et al., 2021), StyleGAN-XL (Sauer et al., 2022), VQGAN (Gu et al., 2022), MobileSwap (Li et al., 2021) and BlendFace (Shiohara et al., 2023). Forensic Small contains $2.78 \times 10^5$ AI-generated images from 4803 generator models and $2.78 \times 10^5$ real images.

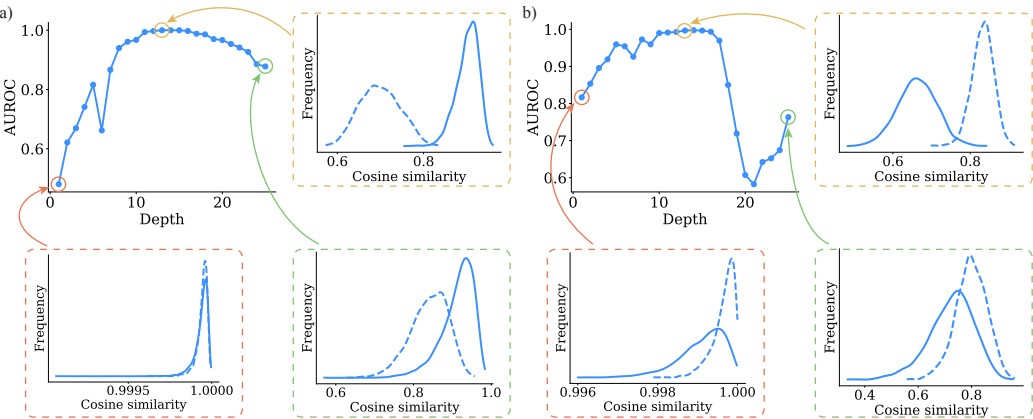

Figure 4: AUROC scores across layers. (a) DF40 DDIM dataset. (b) GenImage BigGAN dataset. Distributions of the cosine similarities between the embeddings of input images and perturbed images for the first layer, intermediate layer and last layer are shown for comparison. Dashed curves are distributions of embeddings of AI-generated images while solid curves are distributions of embeddings of real images. We use the CLIP model to extract features. Elastic transformation is applied for the DDIM dataset and zoom blur for the BigGAN dataset. Severity level 2 is used for both cases.

**Baselines and Metrics**    Both training-based and training-free approaches are selected as baselines to examine the proposed method. For training-based methods, UniDetector (Ojha et al., 2023) uses linear probing on the output of the foundational model to detect AI-generated images. NPR (Tan et al., 2024), based on the observation that up-sampling operations produce generalized forgery artifacts, is an artifact representation approach that captures structural artifacts. AIDE (Yan et al., 2024a) utilizes multiple experts to extract visual artifacts and noise patterns for detecting AI-generated images. SPAI (Karageorgiou et al., 2025) employs the spectral learning to learn the spectral distribution of real images. Generated images are considered out-of-distribution. For training-free methods, RIGID (He et al., 2024) compares the representation similarity between original images and Gaussian noise-perturbed images for detecting AI-generated images. MINDER (Tsai et al., 2024) improves RIGID by contrastive blurring to increase the distance between perturbed embeddings. Aeroblade (Ricker et al., 2024) considers the difference in the difficulty of reconstructing AI-generated and real images and uses it as the detection metric. We evaluate the performance of AI-generated image detection methods using the AUROC score.

## 4.2 COMPARISON WITH BASELINES

Table 1 and Table 2 shows the performance comparison for the AI-generated image detection task. Our approach uses a dataset-dependent configuration while our approach (CC) uses a consistent configuration across the benchmark. The consistent configuration used for the Genimage and Forensic benchmarks is reported in Appendix Section A.2. The configuration is determined by the combination of the randomly sampled training dataset from GenImage and Forensic small benchmarks. The optimal perturbation type, severity level, and intermediate layer are determined by a randomly sampled subset of the training dataset to obtain the performance of using intermediate representations. Our method performs favorably against both training-free and training-based methods.

Model weights of training-based baselines are frozen during the entire inference process. The drawback of training-based methods is the limited generalization to unseen datasets. Training-free methods can generalize well across different datasets but have limited performance. Our method remarkably improves the performance of training-free methods by considering an expanded configuration space.

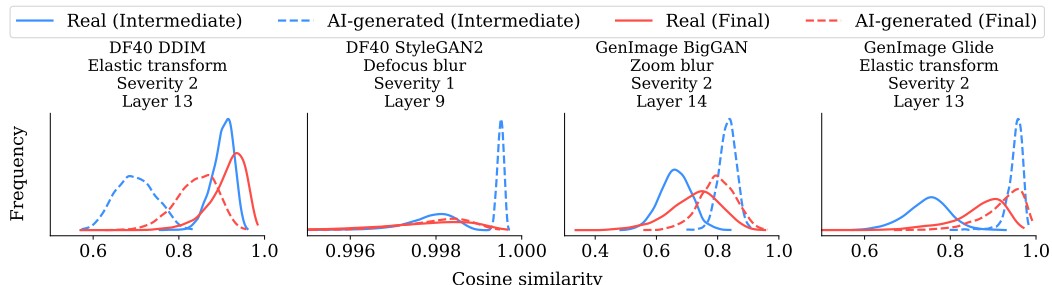

Figure 5: Distribution of cosine similarity between embeddings of AI-generated and real images. Using intermediate layers improves the separation between AI-generated and real images compared to using the last layer. We use the CLIP model as the feature extractor.

### 4.3 INTERMEDIATE LAYER ANALYSIS

Here, we provide a detailed analysis to study the effect of the intermediate layers on AI-generated image detection. We extracted embeddings in all layers (*i.e.* $1 \leq l \leq \ell$). The cosine similarity is computed to predict whether an image is AI-generated as indicated in Equation 2. The AUROC score is used as the metric to examine the prediction performance. Figure 4 shows examples of the AUROC score as a function of model depth. In general, the representations of earlier layers do not provide good separation between real and AI-generated images. While the embedding of the final layer is often used in vision tasks such as image classification, our observation indicates that using an intermediate layer (layers in the middle) in our method usually achieves the optimal detection performance when fixing a perturbation type and a severity level.

In Figure 4, we visualize the distribution of cosine similarity of the first layer, the optimal intermediate layer, and the last layer. Dashed curves correspond to AI-generated images while solid curves correspond to real images. When using the first layer and the last layer, it is difficult to accurately differentiate real and AI-generated images due to the overlap in the distribution. Using intermediate layers, however, improves the separation between distributions of AI-generated and real images.

Figure 5 shows examples of the distribution of cosine similarities for intermediate layers in comparison to final layers. The representations from intermediate layers can yield more separable similarity metrics between real and AI-generated images than the final layers. Hence, using a threshold $\tau$ can well differentiate AI-generated images from real images with the best configuration. We analyze the effect of perturbations (perturbation type and severity level) on the detection performance in Appendix B.1.

## 5 ABLATION STUDY

**Feature extractor**   We examine the performance of our proposed method using different image foundation models as feature extractors. Table 3 shows the performance comparison on the GenImage benchmark and the DF40 benchmark. Instead of the CLIP model (ViT-L/14), when using DINOv2 to extract features, there is a performance degradation. The improvement of the CLIP model over the DINOv2 model can be attributed to the intrinsic dimension analysis in Section 3.3, where we show CLIP has a higher intrinsic dimension than DINOv2, offering more versatile intermediate representations for AI-generated image detection. When using ConvNeXtv2 as the feature extractor, we observe a pronounced performance degradation.

**Subset size**   We use a randomly sampled subset of the training dataset to determine the optimal configuration: intermediate layer, perturbation type, and severity level. We test the effect of different subset sizes on the prediction performance. Figure 6 shows the result on the GenImage benchmark. As the subset size decreases, the prediction performance degrades. We do not observe a significant performance improvement when using a subset that is larger than 30% of the test dataset size.

Table 3: Comparison of using different pretrained image foundation models in our method: DINOv2 (Oquab et al., 2023), CLIP (ViT-L/14) (Radford et al., 2021). and ConvNeXtv2 Woo et al. (2023)

| Foundation model | GenImage benchmark | | | | | | | | |
|---|---|---|---|---|---|---|---|---|---|
| | BigGAN | SD v1.4 | VQDM | ADM | Glide | Midjourney | SD v1.5 | Wukong | Avg |
| AUROC | | | | | | | | | |
| CLIP | 0.9982 | 0.9240 | 0.9475 | 0.9825 | 0.9996 | 0.9031 | 0.9209 | 0.8739 | 0.9437 |
| DINOv2 | 0.9876 | 0.8655 | 0.9466 | 0.9423 | 0.9987 | 0.8416 | 0.8474 | 0.8454 | 0.9094 |
| ConvNeXtv2 | 0.9845 | 0.8079 | 0.8263 | 0.7645 | 0.9703 | 0.7468 | 0.8125 | 0.8146 | 0.8409 |
| AP | | | | | | | | | |
| CLIP | 0.9980 | 0.9118 | 0.9289 | 0.9813 | 0.9996 | 0.9136 | 0.9081 | 0.8569 | 0.9373 |
| DINOv2 | 0.9858 | 0.8202 | 0.9448 | 0.9538 | 0.9988 | 0.8672 | 0.7975 | 0.8246 | 0.8991 |
| ConvNeXtv2 | 0.9696 | 0.7678 | 0.8156 | 0.7696 | 0.9574 | 0.7185 | 0.7774 | 0.7550 | 0.8164 |
| Foundation | DF40 benchmark | | | | | | | | |
| model | DDIM | SiT | StyleGAN2 | StyleGAN3 | StyleGAN-XL | VQGAN | MobileSwap | BlendFace | Avg |
| AUROC | | | | | | | | | |
| CLIP | 0.9998 | 0.9144 | 0.9995 | 1.0000 | 0.8880 | 0.9897 | 0.7066 | 0.9056 | 0.9255 |
| DINOv2 | 0.9904 | 0.8431 | 0.9959 | 1.0000 | 0.9620 | 0.9918 | 0.6402 | 0.9097 | 0.9166 |
| ConvNeXtv2 | 0.9215 | 0.6255 | 0.9751 | 0.9946 | 0.9096 | 0.8710 | 0.5704 | 0.6761 | 0.8180 |
| AP | | | | | | | | | |
| CLIP | 1.0000 | 0.9787 | 0.9999 | 1.0000 | 0.9728 | 0.9979 | 0.944 | 0.9727 | 0.9833 |
| DINOv2 | 0.9981 | 0.9619 | 0.9992 | 1.0000 | 0.9920 | 0.9983 | 0.9281 | 0.9731 | 0.9813 |
| ConvNeXtv2 | 0.9835 | 0.8974 | 0.9950 | 0.9990 | 0.9805 | 0.9733 | 0.9123 | 0.8932 | 0.9543 |

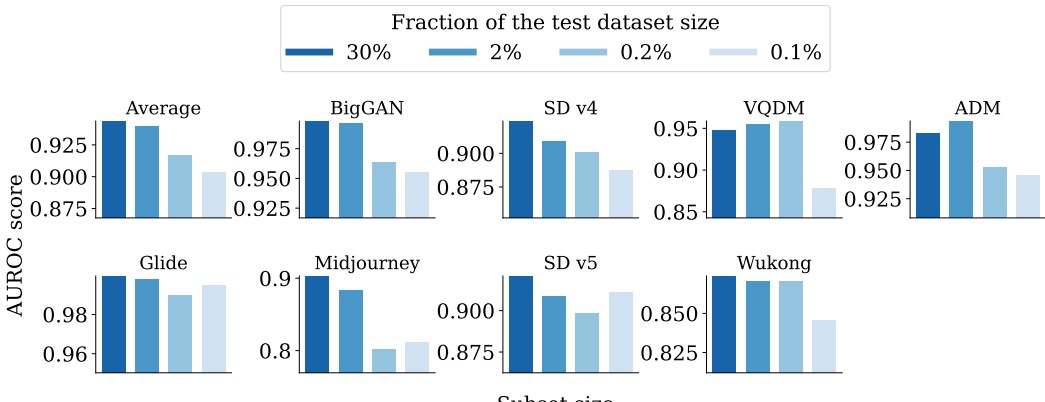

Figure 6: Variation of AUROC score on the GenImage benchmark as a function of different randomly sampled subset sizes. The randomly sampled subset of the training dataset is used to determine the optimal configuration.

## 6 CONCLUSION

In this paper, we propose a novel search-based approach for detecting AI-generated images. By searching for the optimal configuration to obtain the most separable similarity features in the composite space of layer index, perturbation type, and severity level, our approach improves the detection performance over state-of-the-art training-based and training-free methods by a large margin. We also provide comprehensive analysis and intrinsic dimension evaluation to explain how the versatility of the intermediate representations derived from a pretrained image foundation model can be used to design powerful AI-generated image detectors. Our method can be used with any off-the-shelf image foundation model to extract intermediate representations. Hence, we believe the detection performance can scale with the representation learning capability of future image foundation models.

**Ethic Statement** This work focuses on developing a reliable method to address the problem of detecting AI-generated images, with the aim of mitigating risks posed by generative models. Our work can be applied to enhance the reliability of media forensics and support trustworthy information dissemination. The proposed approach does not involve the generation of harmful or offensive content. It employs a publicly available image foundation model without modifying its weights or architecture.

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
