# A  IMPLEMENTATION DETAILS

We use pretrained CLIP to extract features. Besides, we test the performance of using DINOv2 as the feature extractor. Both models use ViT-L/14 as the backbone model. Images are resized to $224 \times 224$ and then used as the input to the foundational vision model.

## A.1  BASELINE IMPLEMENTATION

In RIGID and MINDER baselines, the DINOv2 model is used to detect AI-generated images. We use model weights fine-tuned on the GenImage benchmark in the model inference process for NPR and AIDE baselines. UniDetector trains a classification layer using the curated dataset (Wang et al., 2020). The model weight of SPAI is obtained by training on the curated dataset, where AI-generated images are generated by latent diffusion model (Rombach et al., 2022) while real images are collected from the publicly available dataset (Corvi et al., 2023b).

## A.2  OPTIMAL CONFIGURATION

Using a randomly sampled subset of the training dataset, we are able to determine the optimal configuration, including the intermediate layer index $l$, perturbation function $\epsilon(\cdot)$ and severity level $s$. Table 5 shows the optimal configuration when using DINOv2 as the feature extractor while Table 4 for CLIP as the feature extractor. The performance for using the optimal configuration is shown in Table 1 and Table 2.

The consistent configuration for the GenImage and Forensic Small benchmarks is $l = 13$, perturbation type is defocus blur, severity level is 7 and $\psi(x) = x$. The consistent configuration for the DF40 benchmark is $l = 22$, perturbation type is defocus blur, severity level is 7 and $\psi(x) = 1 - x$.

Table 4: Optimal configuration for detecting AI-generated images using CLIP as the feature extractor.

| Benchmark | Dataset | $s$ | $\epsilon(\cdot)$ | $l$ | Benchmark | Dataset | $s$ | $\epsilon(\cdot)$ | $l$ |
|---|---|---|---|---|---|---|---|---|---|
| GenImage | BigGAN | 2 | Zoom blur | 14 | DF40 | DDIM | 2 | Elastic trans | 13 |
| | SD v1.4 | 7 | Elastic trans | 13 | | SiT | 1 | JPEG compression | 3 |
| | VQDM | 2 | Elastic trans | 13 | | StyleGAN2 | 1 | Defocus blur | 9 |
| | ADM | 3 | Elastic trans | 13 | | StyleGAN3 | 1 | Defocus blur | 9 |
| | Glide | 2 | Elastic trans | 13 | | StyleGAN-XL | 5 | Zoom blur | 11 |
| | Midjourney | 2 | Zoom blur | 13 | | VQGAN | 8 | Impulse noise | 24 |
| | SD v1.5 | 8 | Elastic trans | 13 | | MobileSwap | 5 | Elastic trans | 10 |
| | Wukong | 8 | Elastic trans | 14 | | BlendFace | 4 | Contrast | 10 |

Table 5: Optimal configuration for detecting AI-generated images using DINOv2 as the feature extractor.

| Benchmark | Dataset | $s$ | $\epsilon(\cdot)$ | $l$ | Benchmark | Dataset | $s$ | $\epsilon(\cdot)$ | $l$ |
|---|---|---|---|---|---|---|---|---|---|
| GenImage | BigGAN | 3 | Gaussian noise | 12 | DF40 | DDIM | 8 | JPEG compression | 17 |
| | SD v1.4 | 8 | Contrast | 15 | | SiT | 1 | JPEG compression | 13 |
| | VQDM | 1 | JPEG compression | 24 | | StyleGAN2 | 3 | JPEG compression | 11 |
| | ADM | 5 | Elast transform | 8 | | StyleGAN3 | 3 | JPEG compression | 11 |
| | Glide | 8 | Defocus blur | 5 | | StyleGAN-XL | 1 | JPEG compression | 15 |
| | Midjourney | 1 | JPEG compression | 12 | | VQGAN | 5 | Defocus blur | 24 |
| | SD v1.5 | 7 | Zoom blur | 13 | | MobileSwap | 1 | JPEG compression | 11 |
| | Wukong | 8 | Shot noise | 15 | | BlendFace | 1 | Gaussian noise | 12 |

# B  PERTURBATIONS

Following (Hendrycks & Dietterich, 2019), we apply different perturbation types with different severity levels to input images $\phi : \mathbf{x} \to \epsilon(\mathbf{x})$. Figure 7 shows eight different perturbation types: Gaussian noise, defocus blur, impulse noise, JPEG compression, contrast, shot noise, zoom blur and elastic transform. We use exaggerated severity levels to visualize the effect of different perturbations on original images.

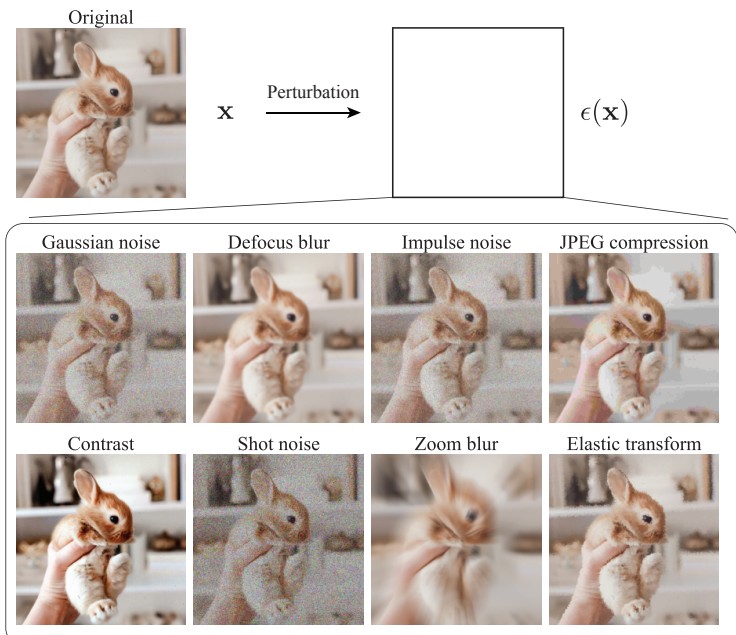

Figure 7: Algorithmically generated corruptions to apply perturbation to input images. Each perturbation type has eight severity levels. We use the cosine similarity between the embeddings of the original image and the perturbed image to make a binary classification on whether the original image is an AI-generated image (*i.e.*, AI-generated image). Perturbations are exaggerated for better visualization purposes.

## B.1 EFFECT OF PERTURBATION

Figure 8 shows the effect of model depth and severity on the detection performance. There is no universal configuration that leads to the best performance. This justifies the practice of using the training dataset to determine the optimal configuration.

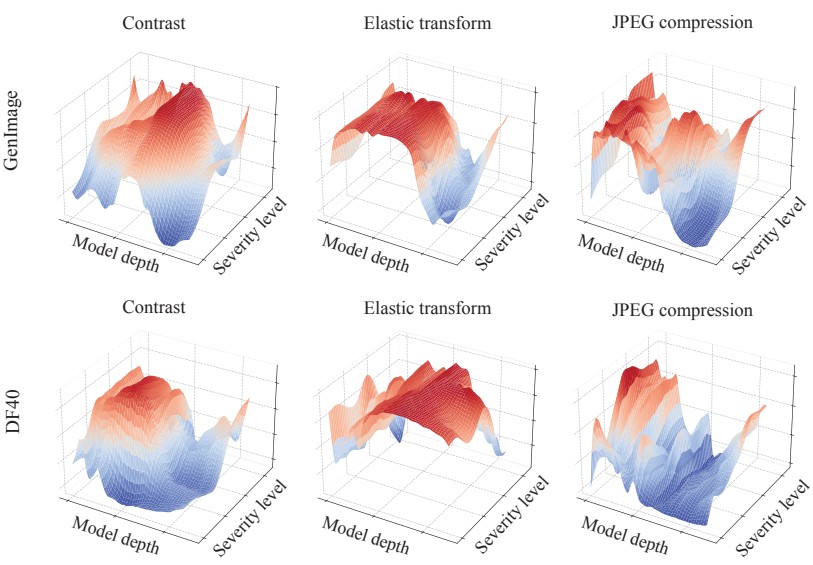

Figure 8: Variation of AUROC score (Z axis) as a function of model depth and severity level for different perturbations on the GenImage benchmark and the DF40 benchmark.

## B.2 SENSITIVITY OF INTERMEDIATE LAYERS TO PERTURBATIONS

Figure 9 shows the profile of cosine similarity between the original and perturbed embeddings. Embeddings are extracted by employing the DINOv2 model as the feature extractor. Similar to the result of using CLIP model shown in Figure 2, in most cases, real images are more robust than AI-generated images. However, there are exceptions such as the Midjourney dataset in the GenImage benchmark.

The training dataset, similar to Figure 2, exhibits a good indicator for the optimal configurations for the test dataset, even though we only use the number of images in the training dataset equal to 30% test dataset size. The optimal configuration, including the best intermediate layer, perturbation type and severity level, is used to detect AI-generated images in the test dataset.

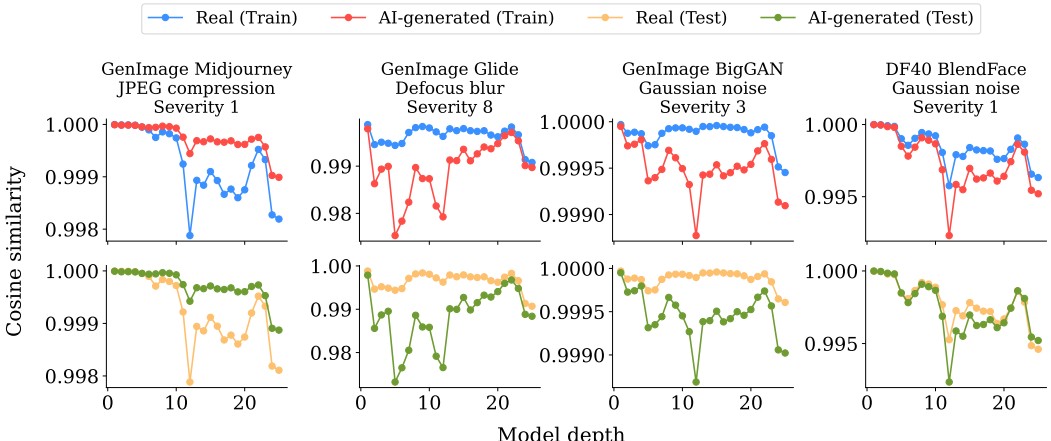

Figure 9: Cosine similarity profile over model depth. We use features extracted by DINOv2 model to compute the cosine similarity. Randomly sampled images in the training dataset with a size of 30% test dataset size are used to represent training dataset in the plot.

## B.3 ROBUSTNESS TO INPUT IMAGE CORRUPTION

Input images are subjected to various corruptions in the real-world scenario. Following the prior works (Wang et al., 2023; Ricker et al., 2024; He et al., 2024), we examine the robustness of the proposed method under three types of perturbations: Gaussian noise, JPEG compression and Gaussian blur. Corruption is applied to the input image $\mathbf{x} \in \mathcal{X}$. After corruption, input images are fed into the feature extractor. Overall, our approach exhibits robustness against perturbations.

Figure 10 shows the robustness of detection approaches on the GenImage benchmark. As the corruption becomes more serious, the nuances of differences between AI-generated and real image features become increasingly intractable. Hence, it becomes increasingly difficult to differentiate real and AI-generated images.

## C PERFORMANCE ON SPECIALIZED DATASET

Table 6 shows the performance on the DF40 benchmark. The consistent configuration used for the DF40 benchmark is different from that for the GenImage and Forensic Small benchmarks. Unlike the GenImage and Forensic Small benchmarks in which there is a variety of image classes, the DF40 benchmark contains merely human faces. Differences in image categories might require a different search configuration.

Figure 11 shows the effect of the training dataset size on the detection performance for the DF40 benchmark. When the fraction reaches 0.2%, there is a pronounced performance drop. Overall, using a small portion of training data is enough to determine the optimal configuration in the search space.

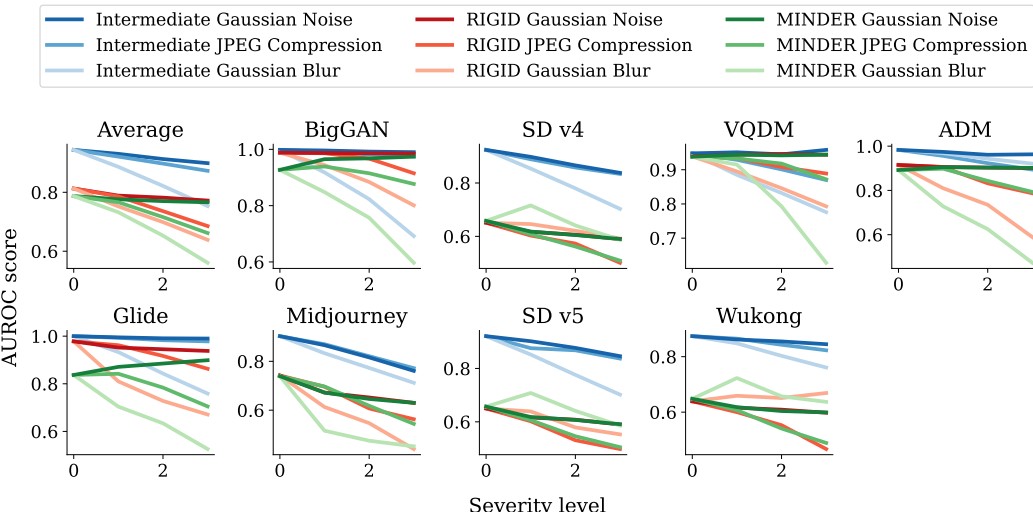

Figure 10: Robustness of AI-generated image detection approaches against three types of corruptions: Gaussian noise, JPEG compression and Gaussian blur.

Table 6: Comparison of AUROC and AP scores on the DF40 (Yan et al., 2024b) benchmark. Ours (CC) uses a consistent configuration across the benchmark.

| Method | Metric | DDIM | SiT | StyleGAN2 | StyleGAN3 | StyleGAN-XL | VQGAN | MobileSwap | BlendFace | Avg |
|---|---|---|---|---|---|---|---|---|---|---|
| | | | | | Training-free method | | | | | |
| AeroBlade | AUROC | 0.5230 | 0.9479 | 0.5337 | 0.7847 | 0.4687 | 0.5021 | 0.3855 | 0.4978 | 0.5804 |
| | AP | 0.4172 | 0.9271 | 0.4125 | 0.8265 | 0.7942 | 0.8613 | 0.8013 | 0.7868 | 0.7284 |
| RIGID | AUROC | 0.8235 | 0.6781 | 0.9217 | 0.9892 | 0.8631 | 0.9494 | 0.5110 | 0.5157 | 0.7815 |
| | AP | 0.9782 | 0.9015 | 0.9802 | 0.9973 | 0.9632 | 0.9888 | 0.8943 | 0.8167 | 0.9400 |
| MINDER | AUROC | 0.9222 | 0.7806 | 0.9144 | 0.9318 | 0.8311 | 0.9930 | 0.5436 | 0.5509 | 0.8085 |
| | AP | 0.9781 | 0.9437 | 0.9752 | 0.9797 | 0.9614 | 0.9985 | 0.9032 | 0.8219 | 0.9452 |
| | | | | | Training-based method | | | | | |
| UniDetector | AUROC | 0.9861 | 0.6596 | 0.9998 | 0.9908 | 0.9310 | 0.9964 | 0.6188 | 0.5761 | 0.8448 |
| | AP | 0.9973 | 0.9135 | 1.0000 | 0.9982 | 0.9862 | 0.9993 | 0.9251 | 0.8369 | 0.9571 |
| NPR | AUROC | 0.9760 | 0.9679 | 1.0000 | 1.0000 | 0.9999 | 0.9973 | 0.3403 | 0.5271 | 0.8511 |
| | AP | 0.9613 | 0.9190 | 0.9999 | 1.0000 | 0.9997 | 0.9932 | 0.8489 | 0.8130 | 0.9419 |
| AIDE | AUROC | 0.6190 | 0.8972 | 0.8894 | 0.9093 | 0.7589 | 0.9120 | 0.5809 | 0.4349 | 0.7502 |
| | AP | 0.8538 | 0.9564 | 0.9656 | 0.9711 | 0.9543 | 0.9729 | 0.8602 | 0.7867 | 0.9151 |
| SPAI | AUROC | 0.4175 | 0.5009 | 0.4425 | 0.5507 | 0.5280 | 0.7374 | 0.4728 | 0.4941 | 0.5180 |
| | AP | 0.7982 | 0.8483 | 0.8212 | 0.8647 | 0.8579 | 0.9361 | 0.8782 | 0.8005 | 0.8506 |
| | | | | | Search-based method | | | | | |
| Ours | AUROC | 0.9998 | 0.9144 | 0.9995 | 1.0000 | 0.8880 | 0.9897 | 0.7066 | 0.9056 | 0.9255 |
| | AP | 1.0000 | 0.9787 | 0.9999 | 1.0000 | 0.9728 | 0.9979 | 0.9440 | 0.9727 | 0.9833 |
| Ours (CC) | AUROC | 0.9667 | 0.8636 | 0.8965 | 0.8369 | 0.8042 | 0.9718 | 0.5580 | 0.6165 | 0.8143 |
| | AP | 0.9923 | 0.9665 | 0.9789 | 0.9630 | 0.9495 | 0.9942 | 0.9040 | 0.8585 | 0.9509 |

# D BASELINE WITH SEARCH

Compared to prior works on robustness of image embeddings, our method requires a training dataset to search for the optimal configuration. To examine the benefit of intermediate layers, we use the same search space excluding optimal layers for the RIGID baseline. In other words, search space for $s$, $\epsilon(\cdot)$ and $\psi(x)$ is the same but only the output embedding is used. The optimal configuration is determined by a randomly sampled subset of the training dataset. Table 7 shows the performance comparison. Our method performs favorably against the RIGID method.

Table 7: Performance comparison when the same configuration search space except for optimal intermediate layers is applied.

| Method | Metric | BigGAN | SD v4 | VQDM | ADM | Glide | Midjourney | SD v5 | Wukong | Avg |
|---|---|---|---|---|---|---|---|---|---|---|
| RIGID (Search) | AUROC | 0.9830 | 0.7819 | 0.8635 | 0.9220 | 0.9569 | 0.8218 | 0.8370 | 0.8290 | 0.8743 |
| | AP | 0.9826 | 0.7649 | 0.8405 | 0.9150 | 0.9530 | 0.7918 | 0.8203 | 0.8117 | 0.8600 |
| Ours | AUROC | 0.9982 | 0.9240 | 0.7649 | 0.9825 | 0.9996 | 0.9031 | 0.9209 | 0.8739 | 0.9437 |
| | AP | 0.9980 | 0.9118 | 0.9289 | 0.9813 | 0.9996 | 0.9136 | 0.9081 | 0.8569 | 0.9373 |

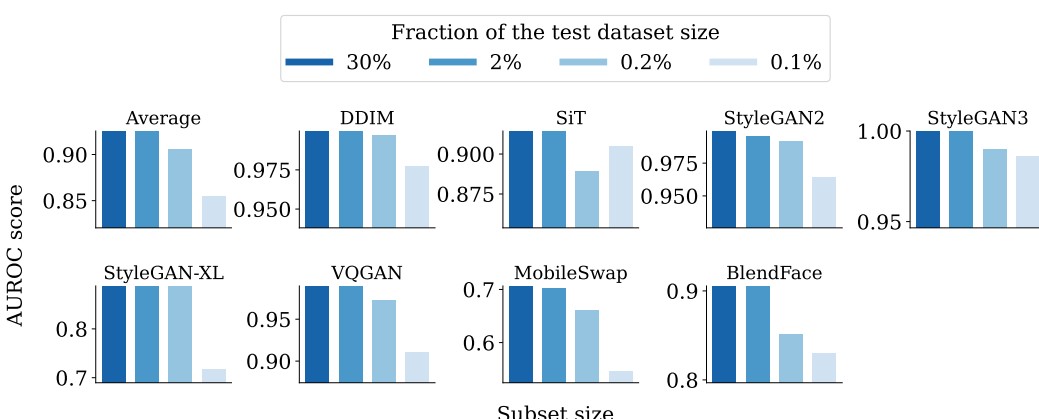

Figure 11: Variation of AUROC score on the DF40 benchmark as a function of different randomly sampled subset sizes. The randomly sampled subset of the training dataset is used to determine the optimal configuration.

## E    AI-GENERATED IMAGE DATASETS

Figure 12 shows examples of AI-generated images in the Forensic Small benchmark. There are $2.78 \times 10^5$ AI-generated images and $2.78 \times 10^5$ real images. Generated images are classified into four categories: GAN, LatDiff, PixDiff and Other. The performance on these four categories is reported in Table 2. To ensure a balanced dataset, we randomly sample real images. The subset of sampled real images has the same size as that of fake images. Comb combines all AI-generated images and all real images.

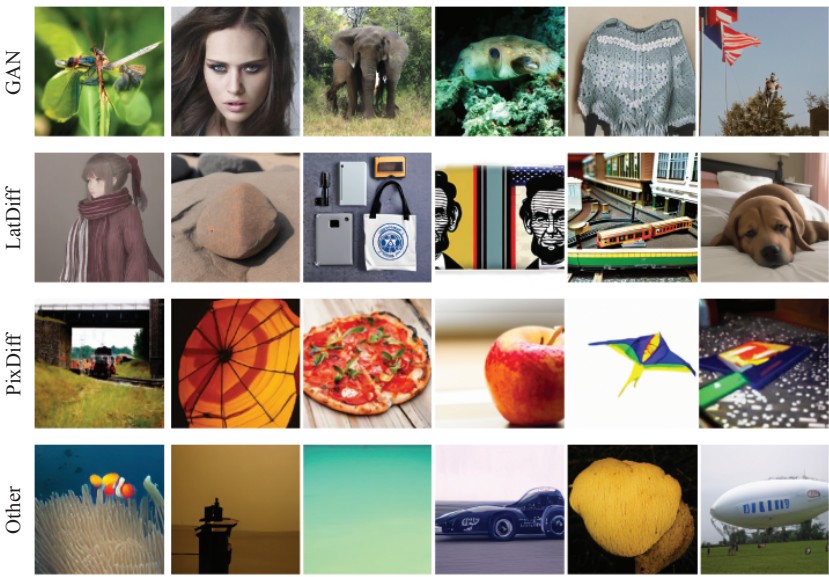

Figure 12: Display of AI-generated images in the Forensic Small benchmark. There are four types of image generators: GAN, LatDiff, PixDiff and other.

Figure 13 shows examples of AI-generated images in the GenImage benchmark. The GenImage benchmark collects more than one million pairs of AI-generated images and retrieved real images. Advanced diffusion models and GAN models are used to produce AI-generated images. 1000 image labels in the ImageNet dataset (Deng et al., 2009) are leveraged to produce AI-generated images.

Figure 14 shows examples of AI-generated images in the DF40 benchmark. The DF40 benchmark uses deepfake techniques to produce AI-generated images including face-swapping, face-reenactment,

entire face synthesis, face editing. Real images are collected from Celeb-DF (CDF) (Li et al., 2020), FFHQ (Karras et al., 2019) and CelebA (Liu et al., 2018). AI-generated images are generated by models of either diffusion model family or GAN family.

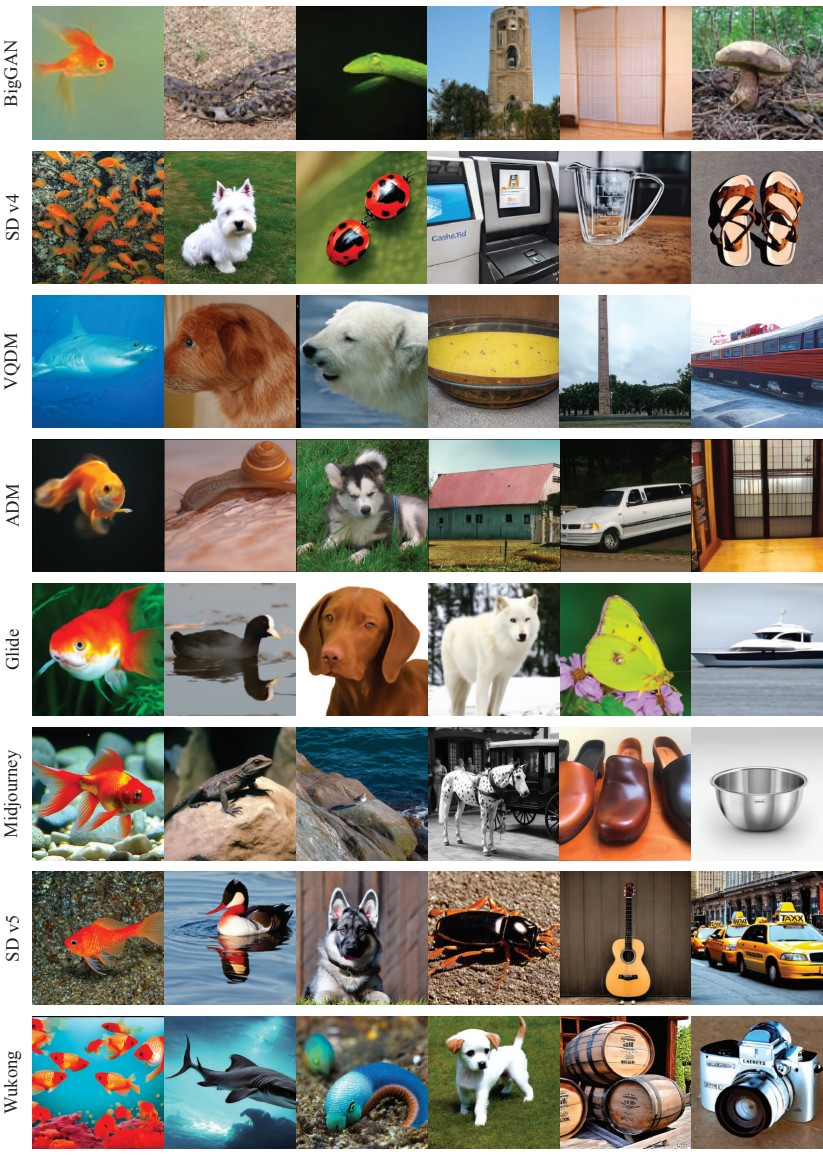

Figure 13: Display of AI-generated images in the GenImage benchmark. Generation models include BigGAN, Stable Diffusion v1.4, VQDM, ADM, GLIDE, Midjourney and Wukong.

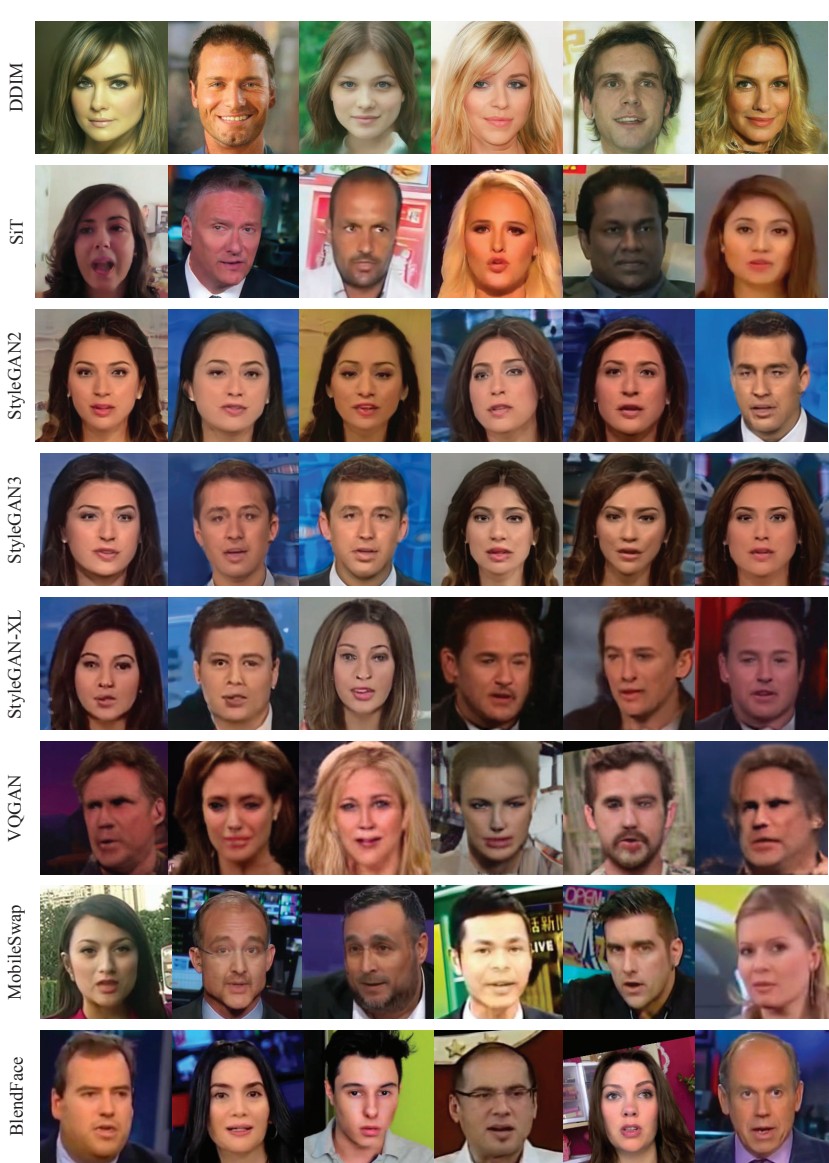

Figure 14: Display of AI-generated images in the DF40 benchmark. Generation models include DDIM, SiT, StyleGAN2, StyleGAN3, styleGAN-XL, VQGAN, MobileSwap and BlendFace.