# OpenReview forum: "Intermediate Representations are Strong Training-Free AI-Generated Image Detectors"
_ICLR.cc/2026/Conference — Submitted to ICLR 2026_

### Official Review · Reviewer_GUBG · 2025-10-21

**Soundness:** 1
**Presentation:** 2
**Contribution:** 2
**Rating:** 2
**Confidence:** 5

**Summary:**

This paper proposes a training-free AI-generated image detection method, which extends previous training-free methods that utilize the difference between the data embedding sensitivity of real and fake images. The proposed method performs a grid search on the configuration space of the intermediate layer, perturbation type, and severity level to identify the best configuration, based on a subset of the training data. Experimental results on GenImage and DF40 show the superiority of the proposed method.

**Strengths:**

1. The analyses on the differences across intermediate layers are interesting.
2. The proposed method is clearly and formally presented.

**Weaknesses:**

1. While the proposed method is claimed to be training-free and to have good generalization performance, it is suspected to be reliant on training data, and the cross-generator generalization is not validated.
  - The selection of the optimal configuration is based on a subset of training data.
  - According to Sec. A.2, the optimal configurations are selected on a **per-generator** basis. In other words, the experiments in this paper assume that the training and test AI-generated images are produced by the same generator. The cross-generator generalization performance is not studied.
  - As suggested by Sec. A.1, the baseline methods are not trained in a per-generator manner. This leads to an unfair comparison with the proposed method.
2. The experiments are conducted solely on GenImage and DF40, where advanced DiT-based image generators such as FLUX and Stable Diffusion 3 are missing. Testing on more challenging and up-to-date datasets such as Chameleon [1] and Community Forensics [2] is important for validating the practical performance of the proposed method.
3. Figure 1(c) is confusing. This 3D mesh plot seems to represent how the "layer" (z-axis) changes along with the severity level and perturbation type, while the color is determined by the detection performance. The "optimal configuration" points to a local minimum. It requires further clarification.
4. The technical contribution is limited as the proposed method only extends previous training-free methods by grid searching on several key factors for an optimal configuration.

[1] A Sanity Check for AI-generated Image Detection. ICLR 2025.
[2] Community Forensics: Using Thousands of Generators to Train Fake Image Detectors. CVPR 2025.

**Questions:**

1. Is the proposed method still effective for the cross-generator scenario? For example, when the training data is limited to one or a few generators, while the test data involves images generated from unseen generators? Can you determine a universal optimal configuration for different test data?
2. Is the proposed method robust to different types of input perturbations?

---

> ### Author Response · Authors · 2025-11-20
>
> We thank the reviewer for thoughtful feedback and constructive comments. We appreciate the time and effort spent evaluating our submission. Below is the one-to-one response:
>
> **Weakness 1**
>
> Please refer to the common reply **2. Clarifying the term "Training-Free"**.
>
> **Weakness 2**
>
> Following the reviewer's suggestion, we conducted experiments to examine the performance of the proposed method on the Community Forensics dataset. Specifically, we used `CommunityForensics-Small` datasets. The performance comparison is reported in *Table 2* of the revised manuscript. We found that using a consistent configuration, our method can achieve better performance than baseline methods.
>
> **Weakness 3**
>
> We thank the reviewer for pointing out this oversight. We modified the plot in *Figure 1*. Please refer to *Figure 1* of the updated manuscript.
>
> **Weakness 4**
>
> Please refer to the common reply **1. Novelty and Contribution**.
>
> **Question 1**
>
> We examined the performance of using a consistent configuration. Please refer to *Table 1* and *Table 2* of the revised manuscript. We found that a consistent configuration can still lead to competitive performance.
>
> **Question 2**
>
> To answer the reviewer's question, we conducted experiments on input corruptions. Please refer to *Figure 10* of the revised manuscript. We found that our method is robust against input corruptions.

---

> > ### Author Response · Authors · 2025-11-25
> >
> > We respectfully invite the reviewer to kindly take a moment to review our responses and updated results. If any part of our clarification remains unclear or requires further explanation, we would be grateful for the feedback. The reviewer's guidance is highly valued, and we sincerely appreciate the reviewer's time and consideration.

---

> > ### Comment · Reviewer_GUBG · 2025-11-27
> >
> > Thank you for the rebuttal, which has addressed some of my concerns. I raise the score from 2 to 4 for now.
> >
> > Here are some further questions and suggestions.
> > 1. My previous major concern is the effectiveness of the proposed method for the cross-generator scenario. The revised "consistent configuration" (CC) results seem strong. However, it is not clearly stated how it is obtained.
> > 2. I recommend testing the performance and comparing it with the baselines on Chameleon (or at least a subset of it), as it consists of high-quality generated images from the online community and filtered by human annotators, close to practical scenarios. With the rapid development of AIGC models, it is important to verify the effectiveness of detectors on high-quality images generated by up-to-date models. It is unclear which models the CommunityForensics-Small dataset covers.
> > 3. I notice that the paper uses notations like "SD v4" and "SD v5", which should be "SD v1.4/1.5" instead.

---

> > > ### Author Response · Authors · 2025-12-02
> > >
> > > We thank the reviewer for increasing the score. We appreciate the time and effort spent evaluating our submission.
> > >
> > > In the revised manuscript, we described how to obtain a consistent configuration (CC):
> > >
> > > > Our approach uses a dataset-dependent configuration while our approach (CC) uses a consistent configuration across the benchmark. The consistent configuration used for the Genimage and Forensic benchmarks is reported in Appendix Section A.2. The configuration is determined by the combination of the randomly sampled training dataset from GenImage and Forensic small benchmarks.
> > >
> > > Besides, we thank the reviewer for pointing out the notations. We modified notations for Stable Diffusion v1.4 and v.1.5. Please refer to the newly revised manuscript.

---

### Official Review · Reviewer_JUDh · 2025-10-27

**Soundness:** 2
**Presentation:** 3
**Contribution:** 3
**Rating:** 4
**Confidence:** 4

**Summary:**

The paper proposes a training-free detector that searches over a configuration space (layer index, perturbation type, severity) and classifies by cosine similarity between an image and its perturbed version using intermediate features from a frozen foundation model (mainly CLIP and DINOv2). Selection of the “best” configuration uses a labeled subset of the training set; inference then applies that single configuration to test images. Claims include sizable AUROC gains on GenImage and DF40 over both training-free (RIGID, MINDER) and several training-based baselines.

**Strengths:**

1. The proposed framework is a simple and code-free adaptation to different backbones (CLIP/DINOv2). The method and Algorithm 1 are easy to reproduce.

2. Intermediate-layer analysis is informative and justifies looking beyond the last layer.

**Weaknesses:**

1. The core paradigm, i.e., comparing feature similarity between an image and a perturbed version, is already central to RIGID and MINDER, which the paper also acknowledges. The method mainly expands the search space (more layers, more perturbations/severities). This reads as an incremental extension rather than a new principle.

2. Stage I requires labeled data and computes similarities across up to 1000+ configurations to pick one (30% of test-set size by default). That is a non-trivial, dataset-specific tuning cost and blurs the “training-free” positioning. Moreover, the paper reports generator-specific best configurations (Tables 3–4), which suggests per-domain tuning. In other words, Tables 3–4 show each generator has a different best (layer, perturbation, severity). In practice, the generator is unknown; tuning per generator (or per image) is infeasible.

3. The paper lacks tests on real capture degradations and unbiased/realistic testbeds such as Chameleon [a] and GenImage-unbiased [b] that specifically probe shortcut/bias failure modes.

[a]A Sanity Check for AI-generated Image Detection
[b] Fake or JPEG? Revealing Common Biases in Generated Image Detection Datasets

4. The paper only focuses on AUROC. While for operational utility and comparison with prior works, please also report Accuracy, Average Precision (AP), which are commonly used in this area.

**Questions:**

1. In terms of Weakness-1:  Please (1) clarify what is fundamentally new beyond search and \psi (Eq. 2), and (2) provide ablations that show why weighted or searched multi-layer choices outperform a strong single-layer baseline under the same corruption budget, and (3) include a “RIGID-with-search” control (same search over severity/types but using only the final layer) to isolate the contribution of intermediate layers.

2. In terms of Weakness-2: (1) Report latency/compute and wall-clock for Stage I vs. a one-pass inference of training-based baselines (e.g., NPR, AIDE) to substantiate the “training cost” motivation. (2) Evaluate a universal configuration chosen on one source set and frozen across unseen generators/datasets (cross-domain transfer), to approximate real deployment.

3. Weaknesses 3-4, I list the questions in the comments.

---

> ### Author Response · Authors · 2025-11-20
>
> We thank the reviewer for thoughtful feedback and constructive comments. We appreciate the time and effort spent evaluating our submission. Below is the one-to-one response:
>
> **Weakness 1**
>
> Please refer to the common reply **1. Novelty and Contribution**.
>
> **Weakness 2**
>
> Please refer to the common reply **2. Clarifying the term "Training-Free"**.
>
> **Weakness 3**
>
> We fully agree on the importance of evaluating the proposed method on unbiased and realistic testbeds. Following the reviewer's suggestion, we conducted experiments to examine the robustness of the proposed method. We applied input corruptions using Gaussian blur, Gaussian noise and JPEG compression. The result is shown in *Figure 10* of the revised manuscript. It demonstrates that our method is robust against input corruptions.
>
> **Weakness 4**
>
> We fully understand the reviewer's concern about the evaluation metric used in our study. Following the reviewer's suggestion, we report AP score in addition to AUROC score. Please refer to *Table 1*, *Table 2* and *Table 3* of the revised manuscript. The result further confirms the effectiveness of our approach in comparison to baseline methods.
>
> **Question 1**
>
> (1) In *Equation 2*, we use the different way to compute $S(\mathbf{x}, \epsilon(\mathbf{x} | s), l)$. We consider the effectiveness of the intermediate representation. As *Figure 3* shows, the intrinsic dimension is highest in the intermediate layer, which indicates that more diverse features are generated. We believe it can better capture the difference in the robustness of image embeddings between real and AI-generated images. Prior works only focus on output embeddings.
>
> (2) A weighted combination of configurations can inevitably increase the computational costs, especially when determining coefficients. By employing a single-layer setting, our approach (using a consistent configuration across benchmarks) can achieve better performance than baseline methods (Please refer to *Table 1* and *Table 2* of the revised manuscript). We believe it is possible that searching over more complex configurations might further boost the performance, but we leave it as future work.
>
> (3) We fully agree that the additional experiment proposed by the reviewer can demonstrate the effectiveness of the intermediate layers. Following the reviewer's suggestion, we use the same search space (except for searching optimal intermediate layer) in our method for the RIGID baseline. The result is shown in *Table 7* of the revised manuscript. We found that there is a pronounced gap between RIGID with search and our method. We thank the reviewer for this great idea to isolate the contribution of the intermediate layers.
>
> **Question 2**
>
> (1) We believe the training cost is reduced because training-based approaches require some iterations of forward and backward propagation. Searching for the optimal layer is an "one-time" cost (due to the one-pass inference) and does not require back propagation, which is typically considered a costly computation.
>
> (2) Following the reviewer's suggestion, we report the performance of applying a universal configuration. Please refer to *Table 1* and *Table 2* of the revised manuscript.

---

> > ### Author Response · Authors · 2025-11-25
> >
> > We respectfully invite the reviewer to kindly take a moment to review our responses and updated results. If any part of our clarification remains unclear or requires further explanation, we would be grateful for the feedback. The reviewer's guidance is highly valued, and we sincerely appreciate the reviewer's time and consideration.

---

### Official Review · Reviewer_EeWq · 2025-10-31

**Soundness:** 3
**Presentation:** 3
**Contribution:** 2
**Rating:** 4
**Confidence:** 5

**Summary:**

This paper proposes a "training-free" method for detecting AI-generated images by applying perturbations, extracting intermediate-layer features, and measuring sensitivity differences between original and perturbed representations to separate real from generated images. The authors evaluate ~1,600 configurations, select the best using limited labeled data, and show on GenImage and DF40 that this approach outperforms existing training-free and some training-based detectors in AUROC.

**Strengths:**

- The method does not require modifying the backbone; it only relies on extracting intermediate-layer representations from vision foundation models, making it straightforward to implement and reproduce.
- The paper is clearly written and logically structured.
- The experimental results are impressive, in some settings even surpassing carefully trained detectors.

**Weaknesses:**

- Existing training-free detectors (e.g., RIGID/MINDER) have already exploited the idea that “real images and synthetic images exhibit different robustness under perturbations.” The main difference in this work is that, instead of examining only the final layer or a single perturbation, it treats the tuple (intermediate layer index × perturbation type × perturbation strength) as a large discrete hyperparameter space, and then searches for the optimal combination. This is an incremental improvement in engineering.
- Although the authors repeatedly emphasize that the method is “training-free,” it still requires labeled images (both real and AI-generated) from approximately 30% of the training set. These images are used to evaluate ~1600 configurations and select the one that yields the best AUROC, which is then deployed for inference.
- It remains unclear how the method behaves when more diverse backbones are considered, and whether visual foundation models with ConvNeXt-style architectures would still be applicable.

**Questions:**

**Typos**:

- In Equation (2), there is an extra closing parenthesis “)”.
- In the caption of Figure 7, “GenImage benchmark” should be “DF40 benchmark”.
- In Table 3 and Table 4, “JPEG compress” should be changed to “JPEG compression”, to be consistent with the main text.

---

> ### Author Response · Authors · 2025-11-20
>
> We thank the reviewer for thoughtful feedback and constructive comments. We appreciate the time and effort spent evaluating our submission. Below is the one-to-one response:
>
> **Weakness 1**
>
> Please refer to the common reply **1. Novelty and Contribution**.
>
> **Weakness 2**
>
> Please refer to the common reply **2. Clarifying the term "Training-Free"**.
>
> **Weakness 3**
>
> Following the reviewer's suggestion, we conducted additional experiments using ConvNeXt-v2-base [1] as the backbone model. The result is shown in *Table 3* of the revised manuscript. We found that the detection performance is inferior compared to transformer-based models.
>
> **Questions 1**
>
> We have modified the typo. Please refer to *Equation 2* of the revised manuscript.
>
> **Questions 2**
>
> We have modified the wrong label. Please refer to *Figure 11* of the revised manuscript.
>
> **Questions 3**
>
> We have modified terms to make them consistent. Please refer to *Table 4* and *Table 5* of the revised manuscript.
>
> [1] Woo, Sanghyun, et al. "Convnext v2: Co-designing and scaling convnets with masked autoencoders." Proceedings of the IEEE/CVF conference on computer vision and pattern recognition. 2023.

---

> > ### Author Response · Authors · 2025-11-25
> >
> > We respectfully invite the reviewer to kindly take a moment to review our responses and updated results. If any part of our clarification remains unclear or requires further explanation, we would be grateful for the feedback. The reviewer's guidance is highly valued, and we sincerely appreciate the reviewer's time and consideration.

---

### Author Response · Authors · 2025-11-20
**Common Reply**

We thank reviewers for their thoughtful feedback and constructive comments. We appreciate the time and effort spent evaluating our submission. Below are common replies to common concerns raised by reviewers:

**1. Novelty and Contribution**

Although prior training-free detectors (e.g., RIGID and MINDER) also rely on perturbations, they operate only on the final-layer embedding (i.e., output embedding) and implicitly assume that real images are always more robust than synthetic ones. Our work challenges both assumptions.

- First, we show that robustness differences are not uniform across depth, but often concentrate in specific intermediate layers. By profiling perturbation responses across all layers, we find that mid-level representations frequently yield stronger separability than the output embedding—revealing a phenomenon that existing detectors systematically miss. This insight has implications well beyond fake-image detection.

- Second, we demonstrate that the direction of robustness is not fixed: real embeddings are not always more stable. Depending on the architecture and generator, the relationship can reverse or vary across perturbation types. To capture this, we introduce $\psi(x)$, which adaptively selects whichever robustness regime (real > fake or fake > real) is most discriminative for each layer–perturbation pair.

Thus, while all methods perturb inputs, our framework shifts the detection principle from “measure output robustness” to “analyze robustness propagation and inversion across depth.” This is not a brute-force expansion, but a new structural hypothesis about the informative patterns hidden within deep networks. Therefore, our detection performance surpasses the baseline methods, as shown in Table 1 and Table 2. We also wish to thank Reviewers JUDh and GUBG for highlighting the strength of intermediate layers analysis.

**2. Clarifying the term "Training-Free"**

We thank the reviewers for highlighting the inconsistency in our use of the term “training-free.” Our original intent was to indicate that the method does not modify model weights or architecture. However, as the reviewers correctly pointed out, our earlier setup did use labeled data to select the optimal layer for each dataset, which is a form of training-like tuning that could make comparisons with strictly training-free baselines unfair and does not assess cross-generator generalization.

To address these concerns, we now clarify the methodological distinction among the different categories of approaches. In our revised manuscript, we distinguish between:

- Training-free approaches, which use no data-driven selection and do not update model parameters

- Training-based approaches, which learn weights or decision functions

- Search-based approach (Our method), which does not modify model weights but does use a configuration determined by the combination of the randomly 30% sampled training dataset of GenImage and the Forensic benchmark.

Therefore, the complexity of our method lies between training-free and training-based methods. Searching for the optimal layer is a “one-time” cost and does not require back propagation. This terminology avoids overclaiming and clarifies that our method is not the same as true training-free detectors.

Importantly, despite this stricter framing, our method still achieves strong performance across benchmarks, consistently surpassing existing baselines and demonstrating the effectiveness of intermediate-layer robustness patterns. Please refer to *Table 1* and *Table 2* in the updated manuscript.

---

### Author Response · Authors · 2025-12-02
**Summary for the Area Chair**

We thank the Area Chair for overseeing the review process. Below is a concise summary of our responses and the clarifications provided in the rebuttal.

**Key Strength Highlighted by Reviewers**

- The analysis of intermediate layers is informative and justifies our approach of examining intermediate layers in lieu of output layers (Reviewers *JUDh* and *GUBG*).
- The proposed method is clearly described (Reviewers *EeWq*, *JUDh* and *GUBG*)

**Major Concerns and Our Response**

- Contribution of our work (Reviewers *EeWq*, *JUDh* and *GUBG*)

We clarified the difference between our method and prior works and highlighted the contribution of our proposed method.

- Terminology of "training-free" (Reviewers *EeWq*, *JUDh* and *GUBG*)

We clarified the terminology of “training-free” – not modify the weight or structure of the pretrained foundational models. We modified the terminology used in the manuscript and highlighted that our method is “search-based”. “Search-based” approach uses the training dataset to determine the optimal configuration, including optimal intermediate layer, perturbation type, severity level and $\psi(x)$.

- Performance on other Dataset (Reviewer *GUBG*)

We examined the performance of the proposed method on the dataset that the reviewer suggested (Forensic-Small). The result indicates the advantage of the proposed method.

**Additional Evidence or Analysis Provided**

During the rebuttal, we added:
- The performance of our proposed method (denoted as Ours (CC)) using a consistent configuration. The result indicates the advantage of the proposed method over both training-free and training-based baselines (Table 1 and Table 2 of the revised manuscript).

- Result on the other dataset (Forensic-Small) suggested by the reviewer *GUBG*.

- Robustness to image corruptions, which indicates the proposed method is robust when there are input image corruptions.

**Conclusion**

We believe that the reviewer’s concerns have been resolved. The reviewer *GUBG* has increased the score and might further increase the score in the second-round discussion. We did not receive the response from the Reviewers *EeWq* and *JUDh*. The paper’s contributions—employing intermediate representations to detect Al-generated images—remain substantial and aligned with the conference’s standards. We respectfully request the Area Chair to consider our clarifications and updated evidence when making the final recommendation.

---

### Meta-Review · Area_Chair_3xt3 · 2026-01-08

**Summary:**

Reviewers’ main concerns were limited novelty beyond prior perturbation based detectors, since the gains may largely come from an expanded labeled configuration search, which also weakens the “training free” claim. They further questioned the practicality and fairness of Stage I tuning and the strength of evidence for realistic cross generator deployment with unknown generators, and requested clearer cost reporting and broader testbeds. The rebuttal improved clarity and added controls, metrics, robustness tests, and universal or consistent configuration results, but key uncertainties remain on whether the contribution goes beyond search and whether deployment realism and compute cost are sufficiently validated. Based on these outstanding issues, I recommend rejection.

**Reviewer Concerns:**

Reviewer EeWq mainly questioned the novelty beyond prior perturbation based training free detectors, the validity of the “training free” framing given the labeled configuration search, and the generality across backbones. The rebuttal partly addressed these points by reframing the method as search based rather than strictly training free, adding a ConvNeXt v2 experiment, and clarifying the claimed conceptual contribution through intermediate layer robustness patterns and potential robustness direction inversion; the main remaining concern is whether the contribution goes beyond an expanded hyperparameter search, namely the novelty issue.
Reviewer JUDh mainly questioned the novelty beyond prior perturbation-based detectors, the practicality of the Stage I labeled configuration search and generator-specific tuning, and the lack of evaluation on unbiased or realistic testbeds, while also requesting additional metrics beyond AUROC. The rebuttal addressed several points by adding a RIGID with search control to isolate the benefit of intermediate layers, reporting AP and corruption robustness results, and providing results under a universal or consistent configuration; the main remaining concerns are whether the contribution goes beyond an expanded search space and the absence of quantitative Stage I compute comparisons.
Reviewer GUBG mainly questioned whether the method truly generalizes in the cross generator setting and whether the comparisons are fair given the training subset based configuration search, while also noting limited dataset coverage, a confusing figure, and potentially incremental novelty. The rebuttal addressed several points by clarifying the search based framing, adding consistent configuration results and an additional evaluation on CommunityForensics Small, providing corruption robustness tests, and fixing presentation and notation issues; the main remaining concerns are whether the approach is convincingly validated under realistic unknown generator deployment and whether the contribution goes beyond an expanded search space.

**Reviewer Scores:**

Reviewer EeWq: 4 → 4. No indication of score change after rebuttal, and the core novelty and “training free” framing concerns likely remain.


Reviewer JUDh: 4 → 4. No follow up response is recorded; the rebuttal addresses several requests but probably does not change the overall view on novelty and deployment realism.

Reviewer GUBG: 2 → 4. This reviewer explicitly raised the score after rebuttal, while keeping additional questions and suggestions.

---

### Decision · Program_Chairs · 2026-01-26

Reject